# SEER: Label-Structured Modality Routing for Multimodal Sentiment Analysis and Intent Recognition

## Abstract

Multimodal sentiment analysis and intent recognition require models to combine textual, acoustic, and visual evidence whose reliability varies across utterances. Adaptive fusion addresses this issue by assigning sample-specific modality weights, but many routing mechanisms estimate reliability from raw features, generic similarity scores, or prototype assignments that are not explicitly tied to the downstream label structure. This paper studies where and how modality confidence should be estimated for adaptive multimodal routing. We first introduce Emotion-Aware Modality Calibration (EAMC), an encoded-space routing method that estimates modality confidence after semantic encoding rather than from raw modality features. We then propose SEER-L1, a label-structured variant of Structured Evidence Estimation and Routing (SEER), that replaces unconstrained prototype matching with modality-private prototypes and shared label anchors, making the routing mechanism more structured and diagnostically interpretable. SEER-L0 is used as a contrastive-only ablation that keeps the EAMC router fixed while adding label-aware contrastive supervision. We evaluate these methods on aligned CMU-MOSI, aligned CMU-MOSEI, and MIntRec, using binary F1 for sentiment analysis and Weighted F1 for intent recognition as the primary metrics. The results show that EAMC provides a strong encoded-routing foundation, while SEER-L1 achieves competitive primary-metric performance with dataset-dependent effects rather than uniform gains over EAMC. Statistical tests, loss ablations, routing diagnostics, and missing-modality analysis further show that the learned weights tend to track lower-error modalities, that prototype supervision improves shared-anchor alignment on ordinal sentiment data, and that all variants remain strongly text-dependent. Overall, the paper presents semantic modality routing as a careful staged investigation: EAMC establishes encoded-space confidence estimation as a practical adaptive-fusion mechanism, and SEER-L1 provides a label-structured refinement that improves interpretability while preserving competitive task performance.

## 1 Introduction

Multimodal sentiment analysis and intent recognition aim to infer affective or communicative states from textual, acoustic, and visual signals. Recent work has moved from explicit fusion operators toward richer multimodal interaction and representation-learning frameworks, including tensorized fusion, low-rank factorization, cross-modal attention, shared-private decomposition, and self-supervised or information-theoretic objectives (Zadeh et al., 2017; Liu et al., 2018; Tsai et al., 2019; Hazarika et al., 2020; Yu et al., 2021; Han et al., 2021; Li et al., 2023). Despite these advances, a central challenge remains: the reliability of each modality varies across utterances. Lexical content may dominate in some examples, while prosody, facial behavior, or cross-modal disagreement may be more informative in others. This sample-dependent variation makes a single static fusion rule poorly matched to sentiment and intent understanding, and motivates adaptive routing mechanisms that estimate modality confidence before fusion.

This challenge has motivated adaptive fusion and expert-routing methods that assign sample-specific modality weights or routing decisions instead of applying the same fusion operation to every utterance (Gao et al., 2024; Fang et al., 2025; Chen et al., 2025). However, the routing signal in many existing approaches is

derived from raw feature statistics, uncertainty estimates, expert activations, or cross-modal agreement patterns. These signals can be useful, but they are not always organized around the downstream sentiment or intent labels. As a result, a high routing score may reflect feature magnitude, modality style, prediction confidence, or agreement with other modalities, rather than the degree to which a modality provides label-relevant evidence for the current sample. This makes adaptive routing difficult to supervise and interpret when modality reliability depends on task-specific semantic content.

We study adaptive routing as a question of representation space: where should modality confidence be estimated, and how should the routing space be structured? Three limitations motivate our formulation. First, routing before semantic encoding can make confidence estimation depend on low-level modality statistics rather than task-level evidence. Second, encoded-space prototype routing is more practical, but unconstrained prototypes may still organize around generic similarity rather than sentiment or intent structure. Third, standard instance-level contrastive objectives such as NT-Xent can be poorly matched to affective labels because they may treat semantically related samples as negatives, even when the samples share the same class or occupy nearby positions on an ordinal sentiment scale (Khosla et al., 2020). These observations motivate a staged investigation of adaptive routing in a space that is semantically encoded and, where appropriate, structured by downstream labels.

We therefore introduce a controlled routing framework centered on two steps. The first step is Emotion-Aware Modality Calibration (EAMC), which moves modality confidence estimation from raw modality features to encoded semantic representations. EAMC uses an unconstrained learnable prototype bank to estimate how strongly each encoded modality representation matches the learned routing space, and then converts these confidence scores into sample-specific modality weights. This provides a practical encoded-space routing baseline and isolates the benefit of estimating reliability after semantic encoding. The second step is Structured Evidence Estimation and Routing (SEER), a label-structured refinement of EAMC. SEER-L1 replaces the free prototype bank with a shared-private prototype structure: modality-specific private prototypes adapt each encoded modality representation, while shared label anchors estimate confidence with respect to the downstream label space. The resulting confidence scores are normalized into routing weights and used in the same weighted-sum fusion rule as EAMC, keeping the encoder family and fusion operation fixed. To isolate the role of label structure, we compare SEER-L1 against two controlled variants. EAMC serves as the encoded-space routing method without explicit label-structured prototypes. SEER-L0 keeps the EAMC router unchanged and adds label-aware contrastive supervision, allowing us to test whether structuring the encoded representations alone is sufficient. This comparison separates three questions: whether routing should be performed after semantic encoding, whether label-aware contrastive geometry changes the routing space, and whether shared-private label anchors provide additional structure for confidence estimation. Extensions that modify temporal evidence extraction or the fusion stage are treated as supplementary appendix variants, since they introduce additional modeling assumptions beyond the main routing question.

We evaluate EAMC, SEER-L0, and SEER-L1 on aligned CMU-MOSI, aligned CMU-MOSEI, and MIntRec (Zadeh et al., 2016; Bagher Zadeh et al., 2018; Zhang et al., 2022). Binary F1 is the primary metric for CMU-MOSI and CMU-MOSEI, and Weighted F1 is the primary metric for MIntRec. Published baselines are retained as external reference points because reporting protocols may differ; direct comparisons focus on local rows evaluated under the same protocol. Beyond downstream metrics, we include loss ablations, statistical tests, routing and prototype diagnostics, missing-modality analysis, and parameter-count analysis. The results show that EAMC provides a strong encoded-space routing foundation, while SEER-L1 preserves competitive primary-metric performance with a more label-structured confidence-estimation mechanism. Statistical tests indicate dataset-dependent differences rather than significant dominance over EAMC, and diagnostics show that SEER-L1 is best interpreted as a structured and analyzable refinement of encoded-space routing rather than as a uniformly dominant performance model.

Our contributions are as follows:

- We study adaptive multimodal fusion as semantic modality routing, where sample-specific modality confidence is estimated from task-relevant encoded representations rather than from raw features or generic fusion statistics.

- We introduce EAMC, an encoded-space routing method that estimates modality reliability through prototype matching over semantic modality representations and uses the resulting confidence scores for weighted fusion.

- We propose SEER-L1, a label-structured extension of EAMC that uses modality-private prototypes and shared label anchors to estimate modality confidence in a more structured routing space.

- We evaluate the framework on CMU-MOSI, CMU-MOSEI, and MIntRec with multi-run comparisons, reproduced adaptive-routing baselines, loss ablations, statistical tests, routing diagnostics, missing-modality analysis, and parameter-count analysis.

The remainder of the paper is organized as follows. Section 2 reviews related work, Section 3 presents EAMC and SEER-L1, Section 4 reports the experimental results and diagnostic analyses, Section 5 discusses the findings and limitations, and Section 6 concludes. Additional ablations, supplementary variants, and reproducibility details are provided in the appendix.

## 2 Related Work

### 2.1 Multimodal sentiment analysis and intent recognition

Multimodal sentiment analysis and intent recognition have developed from explicit fusion operators toward models that capture cross-modal interaction and modality-specific structure. Early methods such as TFN and LMF introduced tensorized or factorized multimodal fusion, while later models used cross-modal attention, shared-private decomposition, self-supervised multi-task learning, mutual-information objectives, and distillation-based strategies (Zadeh et al., 2017; Liu et al., 2018; Tsai et al., 2019; Hazarika et al., 2020; Yu et al., 2021; Han et al., 2021; Li et al., 2023). CMU-MOSI, CMU-MOSEI, and MIntRec are widely used benchmarks for multimodal sentiment and intent understanding (Zadeh et al., 2016; Bagher Zadeh et al., 2018; Zhang et al., 2022). Although these methods improve multimodal representation learning and fusion, many still apply a fixed fusion rule once modality features are computed. We therefore focus on sample-specific modality weighting and on how modality reliability is estimated for adaptive fusion.

### 2.2 Adaptive fusion and mixture-of-experts for multimodal learning

Adaptive routing is related to mixture-of-experts models, which use input-dependent routing to allocate computation across specialized components (Jacobs et al., 1991; Shazeer et al., 2017; Fedus et al., 2022). In multimodal learning, sparse or expert-based routing has been used to encourage modality-sensitive behavior and to improve adaptive fusion (Mustafa et al., 2022; Gao et al., 2024; Fang et al., 2025; Chen et al., 2025). EMOE is the closest comparison because it uses modality-specific dynamic experts for multimodal emotion and intent recognition (Fang et al., 2025). Our distinction lies in the routing signal: rather than estimating confidence from raw features, uncertainty, expert activations, or agreement statistics alone, we examine whether modality confidence should be computed after semantic encoding and, in SEER-L1, further aligned with downstream label structure.

### 2.3 Contrastive supervision and label-structured representation learning

Contrastive learning shapes representation geometry by defining which samples should be close or separated in embedding space. Instance-discrimination objectives such as NT-Xent/InfoNCE treat each instance as its own class, while supervised contrastive learning uses label information to define semantically meaningful neighborhoods (Chen et al., 2020; Khosla et al., 2020). Multimodal contrastive objectives have also been used for cross-modal alignment and large-scale representation learning (Radford et al., 2021; Mustafa et al., 2022). In this work, contrastive supervision serves a narrower role: it structures the modality representations used by the router. This is especially relevant for sentiment regression, where nearby sentiment values should not be treated as hard negatives.

## 2.4 Prototype-based and shared-private modeling

Prototype-based learning organizes an embedding space around learned reference vectors, as in metric-based classification with Prototypical Networks (Snell et al., 2017). Shared-private modeling has also been used to separate task-relevant common structure from modality-specific variation in multimodal learning (Hazarika et al., 2020). Our work connects these ideas at the routing stage. EAMC uses prototype matching to estimate modality confidence in encoded semantic space. SEER-L1 then refines this with a shared-private prototype structure: modality-specific private prototypes adapt each modality, and shared anchors evaluate the adapted representation with respect to the label structure. This design estimates modality reliability in a task-structured routing space rather than through unconstrained prototype matching alone.

## 3 Method

We study adaptive multimodal routing through a controlled progression that keeps the modality encoders, prediction heads, and weighted-sum fusion rule fixed. EAMC estimates modality confidence from encoded semantic representations using an unconstrained prototype bank. SEER-L0 keeps the EAMC router and adds label-aware contrastive supervision to the encoded representations. SEER-L1 replaces the unconstrained prototype bank with shared-private label-structured routing, where modality-private prototypes adapt each modality and shared label anchors estimate confidence with respect to the downstream label space. Extensions that modify temporal evidence extraction or the fusion stage are reported in Appendix D, since they introduce assumptions beyond the main routing mechanism. Figure 1 summarizes the comparison.

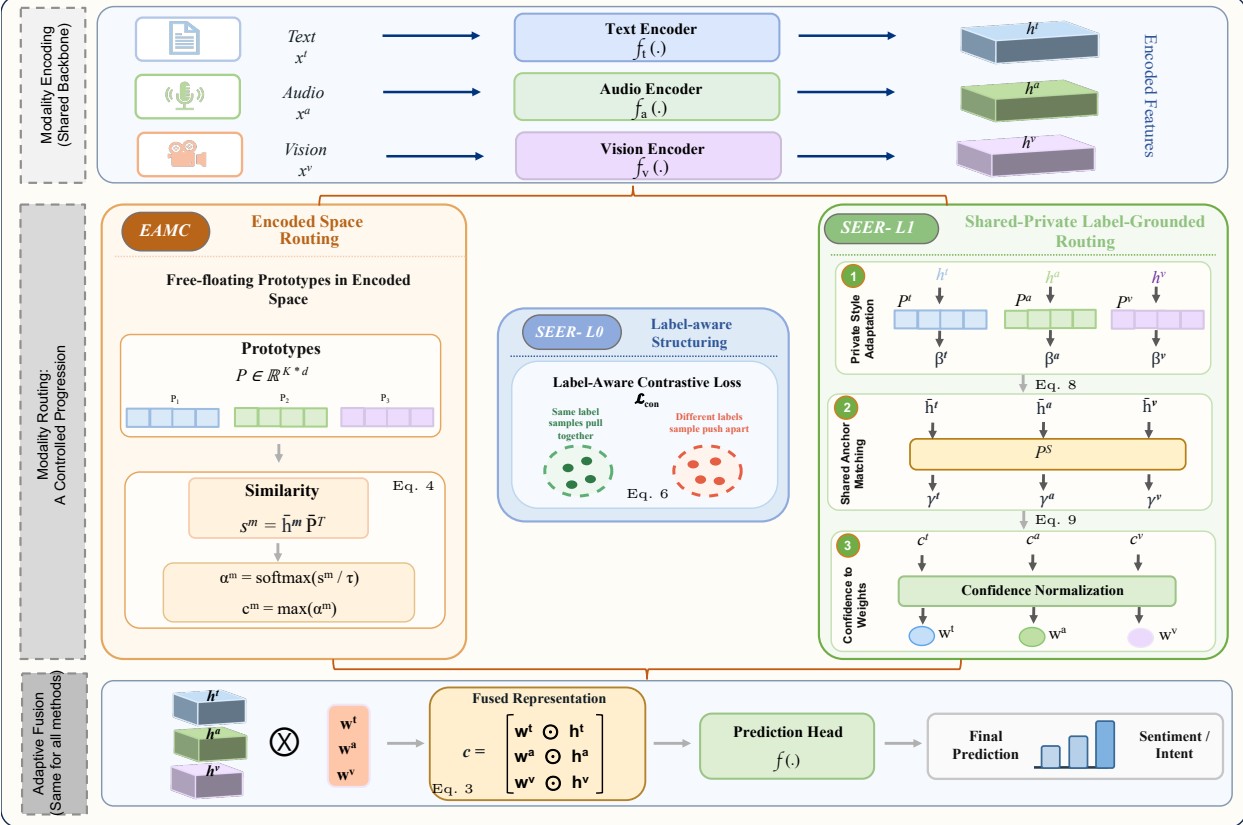

Figure 1: Controlled routing comparison. EAMC estimates modality reliability using an unconstrained encoded-space prototype bank. SEER-L0 keeps this router and adds label-aware contrastive supervision. SEER-L1 replaces the free prototype bank with shared-private label-structured routing, where modality-private prototypes adapt each modality and shared anchors estimate confidence before weighted fusion.

### 3.1 Overview and shared backbone

Let $\mathcal{D} = \{(X_i^t, X_i^a, X_i^v, y_i)\}_{i=1}^N$ denote a multimodal dataset, where $X_i^t$, $X_i^a$, and $X_i^v$ are the text, audio, and visual streams of sample $i$, and $y_i$ is either a sentiment target or an intent label. Let $\mathcal{M} = \{t, a, v\}$ denote the modality set. Each modality is encoded by a modality-specific encoder $\text{Enc}_m(\cdot; \theta_m)$ into a temporal sequence

$$H_i^m = \text{Enc}_m(X_i^m; \theta_m) \in \mathbb{R}^{T_m \times d}, \qquad m \in \mathcal{M}, \tag{1}$$

where $T_m$ denotes the modality-specific sequence length after preprocessing, $d$ is the hidden dimension, and $\theta_m$ denotes the learnable parameters of the modality encoder. In the default path, the sequence is summarized by its final hidden state,

$$h_i^m = H_i^m[T_m] \in \mathbb{R}^d. \tag{2}$$

Each routing method maps the modality representations $\{h_i^m\}_{m \in \mathcal{M}}$ to sample-specific routing weights $\{w_i^m\}_{m \in \mathcal{M}}$, where the weights are nonnegative and normalized across modalities. Given these weights, all controlled variants use the same weighted-sum fusion rule:

$$u_i = \sum_{m \in \mathcal{M}} w_i^m h_i^m. \tag{3}$$

The fused prediction head $q_f(\cdot; \psi_f)$ predicts from $u_i$, while unimodal prediction heads $q_m(\cdot; \psi_m)$ predict from each $h_i^m$. Thus, the controlled comparisons leave the encoder family, unimodal heads, fused head, and fusion rule fixed, and modify only how modality confidence scores and routing weights are computed from $\{h_i^m\}_{m \in \mathcal{M}}$.

### 3.2 Emotion-Aware Modality Calibration (EAMC)

EAMC estimates modality reliability from encoded semantic representations rather than raw modality features. Let $P \in \mathbb{R}^{K \times d}$ be a learnable prototype bank used for similarity-based routing. $P$ is initialized from a standard normal distribution, row-wise $\ell_2$-normalized at initialization, and optimized end-to-end with the rest of the model. During each forward pass, row-wise $\ell_2$-normalization is applied before similarity computation; the raw prototype parameters are not explicitly renormalized after each optimizer update.

For each modality representation $h_i^m$, EAMC computes a soft assignment over the prototype bank using normalized similarity:

$$\alpha_i^m = \text{softmax}\left(\frac{\bar{h}_i^m \bar{P}^\top}{\tau}\right), \qquad c_i^m = \max_k \alpha_{i,k}^m, \tag{4}$$

where $\bar{h}_i^m$ denotes the $\ell_2$-normalized modality representation, $\bar{P}$ denotes the row-wise $\ell_2$-normalized prototype bank, and $\tau$ is a fixed routing-temperature hyperparameter. The confidence score $c_i^m$ is therefore a sample-specific computed quantity indicating the strongest prototype assignment for modality $m$, not a learnable parameter.

The modality confidence scores are normalized across modalities to obtain sample-specific routing weights:

$$w_i^m = \frac{\exp(c_i^m / \tau)}{\sum_{m' \in \mathcal{M}} \exp(c_i^{m'} / \tau)}, \qquad m \in \mathcal{M}. \tag{5}$$

The routed multimodal representation is then computed using the shared fusion rule in Eq. 3.

EAMC therefore isolates the effect of estimating modality reliability after semantic encoding. However, because its prototype bank is not explicitly aligned with sentiment or intent labels, high confidence may reflect strong encoded-space prototype matching without necessarily being tied to downstream label structure. This motivates SEER-L1.

### 3.3 SEER: label-grounded routing

SEER introduces label information into the routing space built by EAMC. We use SEER-L0 as a contrastive-only ablation and SEER-L1 as the main label-structured refinement. SEER-L0 keeps the EAMC router fixed

and adds label-aware contrastive supervision to the encoded modality representations. This tests whether label-aware representation structuring alone is sufficient when the routing mechanism remains unchanged. SEER-L1 changes the confidence-estimation mechanism itself by replacing the free EAMC prototype bank with shared-private label-structured routing.

**Label-aware contrastive structuring (SEER-L0):** SEER-L0 keeps the EAMC router unchanged and augments training with a label-aware contrastive term. Let $z_i^m = g_m(h_i^m; \phi_m)$ denote the output of a modality-specific projection head for modality $m$, where $\phi_m$ denotes learnable projection-head parameters. The contrastive loss is applied independently to each modality branch and then averaged across modalities:

$$\mathcal{L}_{\text{con}} = \frac{1}{|\mathcal{M}|} \sum_{m \in \mathcal{M}} \mathcal{L}_{\text{con}}^m. \tag{6}$$

For a single modality branch, the loss is

$$\mathcal{L}_{\text{con}}^m = -\frac{1}{N} \sum_{i=1}^{N} \sum_{j \neq i} \bar{\omega}_{ij} \log \frac{\exp(\text{sim}(z_i^m, z_j^m)/\tau_c)}{\sum_{k \neq i} \exp(\text{sim}(z_i^m, z_k^m)/\tau_c)}. \tag{7}$$

Here $\text{sim}(\cdot, \cdot)$ denotes cosine similarity, $\tau_c$ is the contrastive temperature, and $\bar{\omega}_{ij}$ determines the label-aware weight assigned to sample $j$ for anchor $i$. For classification, $\bar{\omega}_{ij}$ is uniform over samples with the same label and zero otherwise. For regression, we use soft label-distance weights,

$$\omega_{ij} = \exp\left(-\frac{|y_i - y_j|}{\sigma}\right), \qquad \bar{\omega}_{ij} = \frac{\omega_{ij}}{\sum_{k \neq i} \omega_{ik}},$$

where $\sigma > 0$ controls how quickly the weight decreases with label distance. Thus, nearby sentiment values receive larger positive weights than distant sentiment values. SEER-L0 modifies the training objective but leaves the routing mechanism identical to EAMC.

**Shared-private label-grounded prototype routing (SEER-L1):** SEER-L1 replaces the free EAMC prototype bank with a shared-private structure that separates modality-specific adaptation from label-structured confidence estimation. The private component consists of modality-specific prototype banks, while the shared component consists of anchors used across all modalities to represent the downstream label structure. Let $P_s \in \mathbb{R}^{K_s \times d}$ denote the shared anchor bank, and let $P_{\text{priv}}^m \in \mathbb{R}^{K_p \times d}, m \in \mathcal{M}$, denote modality-specific private prototype banks. For classification, $K_s$ is the number of intent classes and each shared anchor corresponds to one class. For regression, the shared anchors are ordered across the sentiment range and the target anchor is assigned by nearest sentiment value. The private prototype banks are initialized from a standard normal distribution and row-wise $\ell_2$-normalized at initialization. For classification, the shared anchors are also randomly initialized and row-wise normalized. For regression, the shared anchors are initialized as ordered anchors over the sentiment range and then optimized during training. All shared and private prototype parameters are learnable and are updated by backpropagation.

For each modality, SEER-L1 first computes a private prototype assignment and uses it to form a modality-adapted representation:

$$\beta_i^m = \text{softmax}\left(\frac{\bar{h}_i^m (\bar{P}_{\text{priv}}^m)^\top}{\tau}\right), \qquad \tilde{h}_i^m = \beta_i^m P_{\text{priv}}^m. \tag{8}$$

Here $\bar{P}_{\text{priv}}^m$ denotes the row-wise $\ell_2$-normalized private prototype bank used for similarity computation. The adapted representation $\tilde{h}_i^m$ is then matched to the shared label anchors:

$$\gamma_i^m = \text{softmax}\left(\frac{\bar{\tilde{h}}_i^m \bar{P}_s^\top}{\tau}\right), \qquad c_i^m = \max_k \gamma_{i,k}^m. \tag{9}$$

The confidence scores are normalized across modalities using the same routing normalization as EAMC:

$$w_i^m = \frac{\exp(c_i^m/\tau)}{\sum_{m' \in \mathcal{M}} \exp(c_i^{m'}/\tau)}, \qquad m \in \mathcal{M}. \tag{10}$$

The final multimodal representation is computed using the shared fusion rule in Eq. 3.

This routing mechanism changes how confidence scores are estimated while keeping the encoder family, prediction heads, and fusion rule fixed. The private banks adapt each modality into a modality-conditioned prototype space, while the shared anchors convert the adapted representation into a label-structured confidence score. Anchor matching uses $\tilde{h}_i^m$ directly, without a residual connection to $h_i^m$, so SEER-L1 modifies the routing space rather than introducing a new backbone or fusion operator.

To further align the shared anchors with the downstream task, we add prototype supervision. Let $a(y_i)$ denote the target anchor index for sample $i$. For classification, $a(y_i)$ is the intent class label. For regression, $a(y_i)$ is the index of the nearest ordered sentiment anchor. The prototype-supervision loss is

$$\mathcal{L}_{\text{proto}} = \frac{1}{N|\mathcal{M}|} \sum_{i=1}^{N} \sum_{m \in \mathcal{M}} \text{CE}(\gamma_i^m, a(y_i)), \tag{11}$$

where $\text{CE}(\cdot, \cdot)$ denotes the cross-entropy loss.

The prototype loss complements the contrastive term. The contrastive objective encourages label-consistent neighborhoods in the modality representation space, while prototype supervision encourages the shared-anchor assignments to align with the target labels. Supplementary extensions that modify temporal evidence extraction or fusion are described in Appendix D.

### 3.4 Training objective and variant instantiation

All controlled variants use the same encoder family, prediction heads, and weighted-sum fusion rule. Let $\hat{y}_i^f = q_f(u_i; \psi_f)$ denote the fused prediction and $\hat{y}_i^m = q_m(h_i^m; \psi_m)$ denote the unimodal prediction. The fused and unimodal losses are

$$\mathcal{L}_{\text{fuse}} = \frac{1}{N} \sum_{i=1}^{N} \ell(\hat{y}_i^f, y_i), \qquad \mathcal{L}_{\text{uni}} = \frac{1}{N|\mathcal{M}|} \sum_{i=1}^{N} \sum_{m \in \mathcal{M}} \ell(\hat{y}_i^m, y_i),$$

where $\ell$ is the $\ell_1$ loss for CMU-MOSI and CMU-MOSEI and cross-entropy for MIntRec. The full objective is

$$\mathcal{L} = \mathcal{L}_{\text{fuse}} + \mathcal{L}_{\text{uni}} + \lambda_{\text{con}} \mathcal{L}_{\text{con}} + \lambda_{\text{proto}} \mathcal{L}_{\text{proto}} + \mathcal{L}_{\text{shared}},$$

with

$$\mathcal{L}_{\text{shared}} = 0.1 \mathcal{L}_{\text{ent}} + 0.01 \mathcal{L}_{\text{sim}} + 0.1 \mathcal{L}_{\text{ud}}.$$

Here, $\mathcal{L}_{\text{ent}}$ regularizes the modality-weight distribution, $\mathcal{L}_{\text{sim}}$ encourages routing weights to agree with estimated modality usefulness from unimodal prediction errors, and $\mathcal{L}_{\text{ud}}$ encourages consistency between fused and unimodal predictions. These shared terms are fixed across local variants.

EAMC uses the free encoded-space prototype router with $\lambda_{\text{con}} = \lambda_{\text{proto}} = 0$. SEER-L0 keeps the EAMC router and sets $\lambda_{\text{con}} = 0.1$, $\lambda_{\text{proto}} = 0$. SEER-L1 uses the shared-private label-structured router with $\lambda_{\text{con}} = 0.1$ and $\lambda_{\text{proto}} = 0.05$. The learnable components are the modality encoders, prediction heads, projection heads when contrastive supervision is active, and the prototype banks used by the router. The assignment vectors, confidence scores, and routing weights are sample-specific computed quantities rather than learnable parameters.

## 4 Experiments

We evaluate the proposed routing framework on aligned CMU-MOSI and CMU-MOSEI for multimodal sentiment analysis and on MIntRec for multimodal intent recognition. The experiments are designed to assess both task performance and routing behavior. First, we examine whether EAMC provides a competitive encoded-space foundation for adaptive modality routing. Second, we evaluate whether SEER-L1 preserves or improves primary-metric performance when unconstrained prototype routing is replaced with shared-private label-structured routing. Third, we use controlled ablations, statistical tests, routing diagnostics, and missing-modality analysis to examine how the learned confidence scores behave and how much each SEER-specific component contributes.

### 4.1 Datasets, metrics, and implementation

We evaluate on aligned CMU-MOSI and CMU-MOSEI for multimodal sentiment regression, and on MIntRec for multimodal intent classification. The train/validation/test splits are 1,284/229/686 for CMU-MOSI, 16,326/1,871/4,659 for CMU-MOSEI, and 1,334/445/445 for MIntRec. Binary F1 is the primary metric for CMU-MOSI and CMU-MOSEI, while Weighted F1 is the primary metric for MIntRec; secondary metrics are reported in the appendix.

All local results are reported as mean $\pm$ standard deviation over three runs, with each run selected by the best validation primary metric following EMOE (Fang et al., 2025). Published baselines are included as external reference points because feature processing, seed selection, checkpoint selection, and metric implementations may differ. Direct comparisons therefore focus on local rows evaluated with the same feature pipeline, checkpoint-selection rule, and metric implementation. In the main result tables, boldface is applied only within the local rows to avoid implying direct protocol-equivalent comparison with published single-model results.

All local EAMC, SEER-L0, SEER-L1, and reproduced EMOE variants use the same multimodal backbone where applicable. Text is encoded with BERT (Devlin et al., 2019), while audio and video features are projected to a shared hidden dimension and processed by modality-specific Transformer encoders (Vaswani et al., 2017). Exact hyperparameters, seeds, preprocessing details, and reproduction settings are provided in Appendix A.

Table 1: Main results on aligned CMU-MOSI. Published rows are external reference points. Bold indicates the best local result.

| Method | Acc-7 ↑ | Acc-2 ↑ | Binary F1 ↑ | MAE ↓ |
|---|---|---|---|---|
| MulT (Tsai et al., 2019) | 35.1 | 80.0 | 80.1 | 0.936 |
| MISA (Hazarika et al., 2020) | 41.8 | 84.2 | 84.2 | 0.754 |
| Self-MM (Yu et al., 2021) | 45.3 | 84.9 | 84.9 | 0.738 |
| MMIM (Han et al., 2021) | 45.8 | 84.6 | 84.5 | 0.717 |
| DMD (Li et al., 2023) | 46.2 | 83.2 | 83.2 | 0.721 |
| EMOE (published) (Fang et al., 2025) | 47.7 | 85.4 | 85.4 | 0.710 |
| EMOE (reproduced) | 44.71 $\pm$ 2.81 | 85.93 $\pm$ 1.13 | 85.90 $\pm$ 1.11 | 0.731 $\pm$ 0.024 |
| EAMC (ours) | **45.77 $\pm$ 0.25** | 85.62 $\pm$ 0.32 | 85.58 $\pm$ 0.37 | **0.724 $\pm$ 0.011** |
| SEER-L1 (ours) | 44.22 $\pm$ 2.24 | **85.98 $\pm$ 0.61** | **85.96 $\pm$ 0.58** | 0.740 $\pm$ 0.027 |

### 4.2 Main benchmark results

Tables 1-3 report the main benchmark results. Published baselines are included as external reference points, while the local rows provide the controlled comparison among reproduced EMOE, EAMC, and SEER-L1 where available. Across the three datasets, EAMC performs competitively, supporting encoded-space confidence estimation as a strong practical routing foundation. SEER-L1 maintains comparable primary-metric performance while introducing the shared-private label-structured routing mechanism evaluated in later ablations and diagnostics.

Table 2: Main results on aligned CMU-MOSEI. Published rows are external reference points. Bold indicates the best local result.

| Method | Acc-7 ↑ | Acc-2 ↑ | Binary F1 ↑ | MAE ↓ |
|---|---|---|---|---|
| MulT (Tsai et al., 2019) | 52.3 | 82.7 | 82.8 | 0.572 |
| MISA (Hazarika et al., 2020) | 52.3 | 85.3 | 85.1 | 0.543 |
| Self-MM (Yu et al., 2021) | 53.2 | 84.5 | 84.3 | 0.540 |
| MMIM (Han et al., 2021) | 50.1 | 83.6 | 83.5 | 0.580 |
| DMD (Li et al., 2023) | 52.4 | 84.8 | 84.7 | 0.546 |
| EMOE (published) (Fang et al., 2025) | 54.1 | 85.3 | 85.3 | 0.536 |
| EMOE (reproduced) | 52.04 $\pm$ 0.60 | 85.65 $\pm$ 0.13 | 85.62 $\pm$ 0.05 | 0.549 $\pm$ 0.012 |
| EAMC (ours) | **52.52 $\pm$ 0.55** | 85.41 $\pm$ 0.26 | 85.33 $\pm$ 0.22 | **0.543 $\pm$ 0.007** |
| SEER-L1 (ours) | 52.44 $\pm$ 0.50 | **85.66 $\pm$ 0.36** | **85.63 $\pm$ 0.34** | 0.543 $\pm$ 0.011 |

Table 3: Main results on aligned MIntRec. Published rows are external reference points. Bold indicates the best local result.

| Method | Acc ↑ | Weighted F1 ↑ | Precision ↑ | Recall ↑ |
|---|---|---|---|---|
| MAG-BERT (Rahman et al., 2020) | 70.34 | 68.19 | 68.31 | 69.36 |
| MulT (Tsai et al., 2019) | 72.58 | 69.36 | 70.73 | 69.47 |
| MISA (Hazarika et al., 2020) | 72.36 | 70.57 | 71.24 | 70.41 |
| EMOE (published) (Fang et al., 2025) | 72.58 | 70.73 | 72.08 | 70.86 |
| EMOE (reproduced) | 72.36 ± 1.36 | 72.32 ± 1.64 | 73.10 ± 1.87 | 72.36 ± 1.36 |
| EAMC (ours) | **72.88 ± 0.72** | 73.01 ± 0.69 | 74.11 ± 0.93 | **72.88 ± 0.72** |
| SEER-L1 (ours) | 72.73 ± 1.69 | **73.03 ± 1.46** | **74.23 ± 1.03** | 72.73 ± 1.69 |

Across the three benchmarks, the local results show that encoded-space routing provides a strong adaptive-fusion foundation. EAMC is competitive across datasets, while SEER-L1 gives the highest local primary metric on CMU-MOSI, CMU-MOSEI, and MIntRec. The gains are modest, and reproduced EMOE remains close where available, but SEER-L1 preserves competitive task performance while adding a more label-structured routing mechanism. We next examine these differences through controlled ablations, statistical tests, and routing diagnostics.

## 4.3 Controlled routing analysis and statistical tests

We next isolate the effect of the routing design. Table 4 compares the controlled progression from EAMC to SEER-L0 and SEER-L1 using the primary metric for each dataset. EAMC estimates confidence in encoded semantic space using an unconstrained prototype bank. SEER-L0 keeps this router fixed and adds label-aware contrastive supervision, while SEER-L1 replaces the free prototype router with shared-private label-structured routing. This comparison separates the effects of encoded-space routing, label-aware representation structuring, and label-structured prototype routing.

Table 4: Controlled routing ablation on the primary metric. CMU-MOSI and CMU-MOSEI report binary F1, while MIntRec reports Weighted F1. Results are local three-run averages under the protocol in Section 4.1.

| Model | CMU-MOSI Binary F1 ↑ | CMU-MOSEI Binary F1 ↑ | MIntRec Weighted F1 ↑ |
|---|---|---|---|
| EAMC | 85.58 ± 0.37 | 85.33 ± 0.22 | 73.01 ± 0.69 |
| SEER-L0 | 85.49 ± 0.41 | 85.52 ± 0.05 | 72.51 ± 0.85 |
| SEER-L1 | **85.96 ± 0.58** | **85.63 ± 0.34** | **73.03 ± 1.46** |

SEER-L0 produces mixed changes relative to EAMC, suggesting that label-aware contrastive structuring alone does not consistently improve the routing outcome when the confidence-estimation mechanism remains unchanged. SEER-L1 gives the highest mean primary metric in this controlled comparison, but the differences are small relative to run-to-run variation on several datasets. We therefore complement the aggregate results with paired statistical tests for the EAMC–SEER-L1 comparison. This supports using SEER-L0 as a diagnostic ablation rather than as the main proposed variant.

For the primary metric on each dataset, we compare EAMC and SEER-L1 using paired bootstrap confidence intervals and permutation tests over the corresponding local prediction outputs. Table 5 reports the difference as SEER-L1 minus EAMC. These tests characterize the reliability of the observed differences under the local evaluation protocol.

Table 5: Paired statistical comparison between SEER-L1 and EAMC on the primary metric. Differences are reported as SEER-L1 minus EAMC. Positive values favor SEER-L1 and negative values favor EAMC.

| Dataset | Metric | Difference | 95% bootstrap CI | Permutation $p$ |
|---|---|---|---|---|
| CMU-MOSI | Binary F1 | +0.38 | [−0.61, 1.28] | 0.750 |
| CMU-MOSEI | Binary F1 | −0.32 | [−0.81, 0.17] | 0.208 |
| MIntRec | Weighted F1 | +1.74 | [−0.39, 3.63] | 0.082 |

The statistical tests show dataset-dependent differences between EAMC and SEER-L1. On CMU-MOSI, SEER-L1 improves mean binary F1 over EAMC under the strict seed-matched comparison, but the confidence interval overlaps zero. On CMU-MOSEI, the difference is small and slightly favors EAMC under the paired test. On MIntRec, SEER-L1 shows a positive Weighted-F1 trend, although the confidence interval again overlaps zero. These results support interpreting SEER-L1 as a label-structured refinement that preserves competitive task performance, with its added value further examined through loss ablations and routing diagnostics.

## 4.4 Loss ablations

We next isolate the SEER-specific objective terms. Table 6 reports the MIntRec loss ablation because it provides the cleanest completed-run comparison for the full SEER-L1 objective and its loss removals. The full SEER-L1 objective gives the highest mean Weighted F1 and precision among the loss-ablation variants. Removing $\mathcal{L}_{\text{con}}$ produces the largest decrease in Weighted F1, while removing $\mathcal{L}_{\text{proto}}$ has a smaller effect on this dataset. Removing both losses remains competitive but does not exceed the full objective. These results suggest that the auxiliary losses act as useful regularizers for the label-structured router, while their effects are not large enough to treat them as independent performance drivers across all settings.

Table 6: MIntRec loss ablation for SEER-L1. Weighted F1 is the primary metric. Results are local three-run averages under the protocol in Section 4.1.

| Variant | Acc ↑ | Weighted F1 ↑ | Precision ↑ | Recall ↑ |
|---|---|---|---|---|
| EAMC | **72.88 ± 0.72** | 73.01 ± 0.69 | 74.11 ± 0.93 | **72.88 ± 0.72** |
| SEER-L0 | 72.73 ± 0.57 | 72.51 ± 0.85 | 73.25 ± 1.38 | 72.73 ± 0.57 |
| SEER-L1 full | 72.73 ± 1.69 | **73.03 ± 1.46** | **74.23 ± 1.03** | 72.73 ± 1.69 |
| SEER-L1 w/o $\mathcal{L}_{\text{con}}$ | 72.36 ± 0.81 | 72.61 ± 0.77 | 73.90 ± 0.53 | 72.36 ± 0.81 |
| SEER-L1 w/o $\mathcal{L}_{\text{proto}}$ | **72.88 ± 1.28** | 72.96 ± 0.97 | 73.99 ± 0.96 | **72.88 ± 1.28** |
| SEER-L1 w/o both | 72.51 ± 1.11 | 72.85 ± 0.76 | 73.95 ± 0.15 | 72.51 ± 1.11 |

The loss ablation also helps interpret SEER-L0. Label-aware contrastive supervision can shape the representation space, but by itself it does not guarantee stronger routing when the EAMC confidence-estimation mechanism is unchanged. In SEER-L1, the contrastive and prototype losses operate together with the shared-private router: $\mathcal{L}_{\text{con}}$ encourages label-consistent modality representations, while $\mathcal{L}_{\text{proto}}$ directly supervises shared-anchor assignments. An additional CMU-MOSEI loss ablation is provided in Appendix B, where we observe similarly small and dataset-dependent effects.

## 4.5 Routing behavior and robustness diagnostics

We next examine whether the learned confidence scores behave like reliability estimates. For routing reliability, we compare the top-routed modality for each test sample with the modality that has the lowest unimodal prediction error for the same sample. A random router would match the lowest-error modality in approximately one third of samples. We also report the Spearman correlation between routing weights and negative unimodal error, where higher values indicate stronger agreement between assigned confidence and unimodal reliability.

For prototype alignment, we evaluate whether the shared-anchor assignments in SEER-L1 correspond to the label-derived anchor structure. This analysis is most directly interpretable on CMU-MOSEI, where the sentiment labels define an ordinal target space. Anchor accuracy measures exact agreement with the target sentiment anchor, while anchor distance measures the average distance from the target anchor on the ordinal grid.

Table 7 provides mechanism-level evidence for SEER-L1. The full model selects the lowest-error modality in 53.82% of test samples, above the chance level of approximately 33.3%, and shows positive rank correlation between routing weights and negative unimodal error. Removing both SEER-specific losses lowers top-route accuracy, Spearman correlation, and anchor accuracy, while increasing anchor distance from 1.069 to 1.969. Removing only $\mathcal{L}_{\text{proto}}$ produces a larger degradation in prototype alignment, reducing anchor accuracy from

Table 7: Routing and prototype diagnostics on aligned CMU-MOSEI. Top-route accuracy measures how often the highest-weighted modality is also the modality with the lowest unimodal prediction error. Spearman measures rank correlation between routing weights and negative unimodal error. Anchor accuracy and anchor distance evaluate agreement with the label-derived ordinal sentiment anchor.

| Variant | Top-route Acc. ↑ | Spearman ↑ | Anchor Acc. ↑ | Anchor Dist. ↓ |
|---|---|---|---|---|
| SEER-L1 | **53.82 ± 0.20** | **0.197 ± 0.056** | **32.54 ± 2.84** | **1.069 ± 0.175** |
| SEER-L1 w/o both | 51.91 ± 0.85 | 0.162 ± 0.040 | 17.40 ± 0.49 | 1.969 ± 0.130 |
| SEER-L1 w/o $\mathcal{L}_{proto}$ | 47.49 ± 10.04 | 0.126 ± 0.079 | 16.34 ± 1.36 | 2.152 ± 0.187 |

32.54% to 16.34% and increasing anchor distance from 1.069 to 2.152. These results support the intended interpretation of SEER-L1 as a routing mechanism whose confidence scores tend to reflect modality reliability and whose shared anchors are shaped by prototype supervision. These diagnostics do not make the routing weights causal explanations, but they show that the learned confidence scores are empirically related to unimodal reliability.

We also evaluate missing-modality behavior by removing one modality at test time and measuring the change in the primary metric relative to the full-input setting. This analysis is not intended to show robustness dominance by any single variant; instead, it characterizes how strongly each routing method depends on text, audio, and vision under controlled perturbations.

Table 8: Missing-modality analysis. Values report the change in the primary metric relative to the full-input setting. CMU-MOSI and CMU-MOSEI use binary F1; MIntRec uses Weighted F1. More negative values indicate larger degradation after removing the modality.

| Dataset | Model | No text | No audio | No video |
|---|---|---|---|---|
| CMU-MOSI | EAMC | -29.41 | -0.61 | -1.19 |
| CMU-MOSI | SEER-L0 | -46.39 | -0.38 | -1.00 |
| CMU-MOSI | SEER-L1 | -44.90 | +0.10 | -0.45 |
| CMU-MOSEI | EAMC | -35.75 | -2.88 | -0.11 |
| CMU-MOSEI | SEER-L0 | -29.50 | -0.87 | -0.02 |
| CMU-MOSEI | SEER-L1 | -35.79 | -1.29 | -0.42 |
| MIntRec | EAMC | -67.88 | -0.09 | -0.06 |
| MIntRec | SEER-L0 | -68.05 | +0.06 | -0.05 |
| MIntRec | SEER-L1 | -69.62 | -0.08 | -0.21 |

Table 8 shows that all variants remain strongly text-dependent. Removing text causes large drops across all three datasets, especially on MIntRec, while removing audio or video has much smaller effects. This pattern is consistent with the language-centered nature of the evaluated benchmarks. The analysis therefore provides transparency about modality dependence and shows that SEER-L1 should not be interpreted as uniformly more robust to missing modalities than EAMC.

## 4.6 Parameter efficiency

We also compare trainable parameter counts under the local implementations. The comparison is implementation-specific: in our reproduced EMOE implementation, the raw-feature routing module scales with flattened input dimensionality, whereas EAMC and SEER-L1 estimate modality confidence in encoded feature space. Under the same counting rule, SEER-L1 uses 11.2M non-BERT trainable parameters on CMU-MOSI and CMU-MOSEI, compared with 207.8M and 251.7M for reproduced EMOE. On MIntRec, the reproduced EMOE count is specific to the local feature configuration, where the raw-feature router alone contains 1003.6M trainable parameters. EAMC and SEER-L1 have similar parameter counts because they share the same backbone and differ mainly in the small prototype-routing structure. Exact trainable parameter counts and the performance-efficiency visualization are provided in Appendix E.

Additional routing diagnostics, supplementary architectural extensions, unaligned benchmark results, and qualitative representation visualizations are provided in the appendix. We treat the qualitative visualizations as illustrative and use the quantitative ablations, statistical tests, missing-modality analysis, and routing diagnostics as the main evidence for interpreting the routing mechanism.

## 5   Discussion and Limitations

The experiments support a focused interpretation of SEER-L1 as a label-structured refinement of encoded-space modality routing. EAMC shows that estimating modality reliability after semantic encoding is already a strong adaptive-fusion design, while SEER-L0 shows that adding label-aware contrastive supervision alone does not consistently improve routing when the confidence-estimation mechanism remains unchanged. SEER-L1 changes this mechanism by replacing unconstrained prototype matching with shared-private label-structured routing. Under the local protocol, SEER-L1 remains competitive with reproduced EMOE where available and achieves the highest mean local primary metric on CMU-MOSI, CMU-MOSEI, and MIntRec, although the statistical tests show that these margins should be interpreted cautiously.

The ablations and diagnostics clarify why the gains are moderate but interpretable. The loss ablations show that $\mathcal{L}_{con}$ and $\mathcal{L}_{proto}$ act as dataset-dependent regularizers rather than uniformly dominant performance drivers. The routing diagnostic shows that SEER-L1 weights tend to favor lower-error modalities, while the CMU-MOSEI prototype diagnostic shows that $\mathcal{L}_{proto}$ improves alignment with label-derived ordinal anchors. Together, these findings suggest that the main value of SEER-L1 is a more constrained evidence space for estimating modality confidence, rather than simply increased routing flexibility.

This work has several limitations. First, the primary-metric gains are modest, and the paired statistical tests do not show significant dominance over EAMC under the current three-run local protocol. Second, the experiments focus mainly on English benchmarks with aligned multimodal features, which limits conclusions about multilingual, asynchronous, noisy, or naturally corrupted multimodal settings. Third, the routing weights should not be interpreted causally: a high routing weight indicates stronger alignment with the learned routing space, not proof that a modality uniquely caused the prediction. Finally, supplementary extensions such as prototype-attentive temporal evidence extraction and relation-aware fusion are reported in the appendix because they introduce assumptions beyond the main routing mechanism and do not consistently improve the primary results.

Future work should evaluate label-structured routing on multilingual data, asynchronous or corrupted modalities, and broader sentiment, emotion, and intent datasets. Another direction is to design anchor structures that better preserve ordinal sentiment geometry or uncertainty-aware intent relationships during training. More generally, label-structured routing could be extended beyond static modality weighting toward temporal evidence selection, calibrated uncertainty estimation, and explanation-oriented multimodal reasoning.

## 6   Conclusion

This paper examined adaptive multimodal routing for sentiment analysis and intent recognition, focusing on where and how modality confidence should be estimated before fusion. We first introduced EAMC as an encoded-space routing method that estimates modality reliability from semantic modality representations. We then proposed SEER-L1, a shared-private label-structured routing method that uses modality-private prototypes and shared label anchors to estimate sample-specific modality confidence. Controlled comparisons with EAMC and SEER-L0 show that SEER-L1 provides competitive primary-metric performance under a multi-run protocol, while ablations and diagnostics indicate that label-aware contrastive supervision and prototype supervision provide dataset-dependent but interpretable benefits. The results suggest that label-structured confidence estimation is a useful mechanism for making adaptive multimodal fusion more structured and analyzable, even when its empirical gains are moderate and vary across datasets and metrics.

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

## A    Reproducibility Details

All local EAMC, SEER, and reproduced EMOE experiments use the feature pipeline, metric implementation, and per-run checkpoint-selection rule described in Section 4.1. For CMU-MOSI and CMU-MOSEI, the primary metric is binary F1. For MIntRec, the primary metric is Weighted F1. Each local result is reported as mean $\pm$ standard deviation over three runs. For CMU-MOSI, we use the strict seed-matched set $1111, 2222, 4444$ for the EAMC–SEER-L1 comparison reported in the main text. Unless otherwise stated, local models use hidden size 256, four Transformer layers, eight attention heads, and batch size 16. EAMC uses eight encoded-space prototypes with routing temperature 0.07. SEER-L1 uses the same routing temperature for private and shared-anchor matching.

Table 9: Local evaluation protocol used for the main controlled comparisons.

| Dataset | Primary metric | Seeds | Main-text use |
|---|---|---|---|
| CMU-MOSI | Binary F1 | $1111, 2222, 4444$ | Main results and statistical test |
| CMU-MOSEI | Binary F1 | $1111, 2222, 3333$ | Main results and statistical test |
| MIntRec | Weighted F1 | $1111, 2222, 3333$ | Main results and statistical test |

Published baselines are treated as external reference points because they may differ in feature processing, training details, seed convention, checkpoint-selection practice, and metric implementation. Direct comparisons in the main text are therefore based on local rows evaluated under the same reporting rule. Parameter counts are generated using the same counting script and rule for reproduced EMOE, EAMC, and SEER-L1. Additional loss ablations, diagnostics, supplementary model variants, and qualitative visualizations are reported in the following appendix sections.

## B    Additional Loss Ablations

This section reports additional loss ablations to complement the main MIntRec ablation in Section 4.4. The ablations evaluate the contribution of label-aware contrastive supervision, $\mathcal{L}_{\mathrm{con}}$, and prototype supervision, $\mathcal{L}_{\mathrm{proto}}$, under the same local reporting format used in the main experiments.

We report the CMU-MOSEI ablation because it provides a stable supplementary sentiment benchmark for examining the SEER-L1 objective terms. CMU-MOSI is not repeated in this appendix because the main strict seed-matched MOSI comparison is already summarized through the controlled routing and statistical analyses in Section 4.3.

Table 10:   Additional loss ablation on CMU-MOSEI. The table complements the MIntRec loss ablation in the main text and shows the dataset-dependent effect of $\mathcal{L}_{\mathrm{con}}$ and $\mathcal{L}_{\mathrm{proto}}$.

| Variant | Acc-7 $\uparrow$ | Acc-2 $\uparrow$ | Binary F1 $\uparrow$ | MAE $\downarrow$ |
|---|---|---|---|---|
| EAMC | $51.63 \pm 1.79$ | $85.68 \pm 0.33$ | $85.59 \pm 0.36$ | $0.5541 \pm 0.0203$ |
| SEER-L0 | $52.65 \pm 0.67$ | $85.58 \pm 0.29$ | $85.58 \pm 0.26$ | $0.5421 \pm 0.0018$ |
| SEER-L1 | $52.89 \pm 0.62$ | $85.41 \pm 0.53$ | $85.37 \pm 0.45$ | $0.5410 \pm 0.0053$ |
| SEER-L1 w/o $\mathcal{L}_{\mathrm{con}}$ | $53.06 \pm 0.85$ | $85.45 \pm 0.25$ | $85.43 \pm 0.27$ | $0.5415 \pm 0.0045$ |
| SEER-L1 w/o $\mathcal{L}_{\mathrm{proto}}$ | $53.29 \pm 0.29$ | $85.17 \pm 0.24$ | $85.15 \pm 0.17$ | $0.5378 \pm 0.0055$ |
| SEER-L1 w/o both | $52.21 \pm 0.01$ | $85.60 \pm 0.19$ | $85.54 \pm 0.15$ | $0.5407 \pm 0.0027$ |

The CMU-MOSEI ablation shows small and mixed differences across the SEER-L1 loss variants. Removing $\mathcal{L}_{\mathrm{con}}$ or $\mathcal{L}_{\mathrm{proto}}$ does not produce a large degradation in binary F1, although $\mathcal{L}_{\mathrm{proto}}$ is still useful for the prototype-alignment behavior analyzed in Section 4.5. This supports the main-text interpretation that the SEER-specific losses act as dataset-dependent regularizers and diagnostic structuring terms, rather than as uniformly dominant performance drivers.

## C  Full Routing and Prototype Diagnostics

This section provides additional diagnostics for interpreting the learned routing weights and shared-anchor assignments. The main text reports the CMU-MOSEI diagnostic because ordinal sentiment labels make the prototype-alignment analysis directly interpretable. Here, we include additional routing-reliability diagnostics for MIntRec and clarify how the prototype diagnostics should be interpreted across tasks.

**Routing reliability on MIntRec:**  Table 11 reports the MIntRec routing-reliability diagnostic. Top-route accuracy measures how often the highest-weighted modality is also the modality with the lowest unimodal prediction error for the same test sample. With three modalities, a random router would match the lowest-error modality in approximately one third of samples. Pearson and Spearman correlations compare routing weights with negative unimodal error, where larger values indicate stronger agreement between routing confidence and unimodal reliability.

Table 11:  Additional routing-reliability diagnostics on MIntRec. Top-route accuracy measures agreement between the highest-weighted modality and the lowest-error unimodal branch. Pearson and Spearman correlations compare routing weights with negative unimodal error.

| Variant | Top-route Acc. ↑ | Pearson ↑ | Spearman ↑ |
|---------|------------------|-----------|------------|
| SEER-L0 | **84.49 $\pm$ 2.76** | **0.617 $\pm$ 0.035** | 0.522 $\pm$ 0.025 |
| SEER-L1 | 83.00 $\pm$ 2.81 | 0.567 $\pm$ 0.069 | **0.526 $\pm$ 0.042** |

The MIntRec diagnostics show strong agreement between routing weights and unimodal reliability for both SEER-L0 and SEER-L1. Both variants select the lowest-error modality far above the approximate chance level of 33.3%, and both show positive correlations between routing confidence and negative unimodal error. SEER-L0 gives slightly higher top-route accuracy and Pearson correlation, while SEER-L1 gives a slightly higher Spearman correlation. This suggests that the routing weights are meaningfully related to modality reliability on MIntRec, while the label-structured SEER-L1 router remains comparable to the contrastive-only SEER-L0 router on this diagnostic.

**Prototype-label alignment:**  Prototype-label alignment is most directly interpretable for ordinal sentiment datasets, where the shared anchors correspond to ordered sentiment targets. The CMU-MOSEI prototype diagnostic in the main text shows that $\mathcal{L}_{\text{proto}}$ improves shared-anchor agreement with label-derived ordinal anchors. For MIntRec, the intent labels are unordered, so anchor-distance diagnostics are less directly tied to a natural label geometry. We therefore use MIntRec primarily for routing-reliability analysis and use CMU-MOSEI as the main prototype-alignment diagnostic.

## D  Additional Extensions and Supplementary Results

This section reports supplementary architectural extensions and additional benchmark settings. The main text focuses on EAMC, SEER-L0, and SEER-L1 because these variants directly isolate the central routing questions: whether modality confidence should be estimated in encoded semantic space, whether label-aware contrastive structuring changes the routing behavior, and whether shared-private label-structured routing provides a useful confidence-estimation mechanism. The extensions reported here explore additional temporal, fusion, and robustness-oriented mechanisms. They are included to document the broader design space, but they are not used as the primary evidence for the paper's main claims.

Unless otherwise noted, the tables in this section use the same per-run checkpoint-selection and aggregation pipeline described in Section 4.1.

### D.1  Additional extensions

Beyond the EAMC $\rightarrow$ SEER-L0 $\rightarrow$ SEER-L1 comparison, we evaluate three supplementary extensions. SEER-L2 introduces prototype-attentive temporal evidence extraction, SEER-L3 introduces relation-aware

expert fusion, and SEER-L4 introduces robustness-oriented modality dropout. These variants test whether the label-structured routing framework benefits from additional temporal extraction, fusion specialization, or robustness-oriented training.

**Prototype-attentive temporal evidence extraction (SEER-L2):** SEER-L2 replaces the final-state modality representation with prototype-guided temporal evidence extracted from the full modality sequence. This extension tests whether shared anchors can support temporal evidence selection in addition to confidence estimation. In the current experiments, SEER-L2 does not consistently improve over SEER-L1, suggesting that the added temporal attention stage is not reliably beneficial under the evaluated benchmark settings.

**Relation-aware expert fusion (SEER-L3):** SEER-L3 replaces the single weighted-fusion rule with a relation-aware expert mixture. It conditions fusion on pairwise agreement statistics and routing confidences, and predicts weights over consensus, complement, and conflict branches. This extension explores whether structured cross-modal agreement and disagreement can be modeled more explicitly at the fusion stage.

**Robustness-oriented modality dropout (SEER-L4):** SEER-L4 applies modality dropout during training to encourage the model to redistribute confidence when one input stream is missing or unreliable. In this appendix, SEER-L4 is reported under the same standard benchmark evaluation as the other variants. A broader robustness study involving naturally corrupted, asynchronous, or systematically missing modalities is left to future work.

## D.2 Supplementary architecture of SEER-L3

SEER-L3 is a supplementary extension that replaces the single weighted-fusion rule with a relation-aware expert mixture. While EAMC, SEER-L0, and SEER-L1 use the same weighted-sum fusion rule, SEER-L3 introduces an additional fusion-stage router that conditions on cross-modal agreement statistics and modality-confidence scores. This extension is included to document the broader SEER design space, but it is not used as the main evidence for the label-structured routing claims.

For each sample $i$, SEER-L3 computes pairwise agreement scores between the normalized modality representations:

$$\rho_i^{ta} = (\bar{h}_i^t)^\top \bar{h}_i^a, \qquad \rho_i^{tv} = (\bar{h}_i^t)^\top \bar{h}_i^v, \qquad \rho_i^{av} = (\bar{h}_i^a)^\top \bar{h}_i^v. \tag{12}$$

These agreement scores are summarized by their mean and maximum:

$$\mu_i = \frac{1}{3} \left( \rho_i^{ta} + \rho_i^{tv} + \rho_i^{av} \right), \qquad \nu_i = \max \left\{ \rho_i^{ta}, \rho_i^{tv}, \rho_i^{av} \right\}. \tag{13}$$

Let $\mathbf{c}_i = [c_i^t, c_i^a, c_i^v]$ denote the modality-confidence vector. The SEER-L3 router uses

$$r_i = [\mu_i, \nu_i, c_i^t, c_i^a, c_i^v] \tag{14}$$

as input and predicts mixture weights over consensus, complement, and conflict experts:

$$\boldsymbol{\pi}_i = \text{softmax}(\text{MLP}(r_i)), \qquad \boldsymbol{\pi}_i = [\pi_{i,\text{cons}}, \pi_{i,\text{comp}}, \pi_{i,\text{conf}}]. \tag{15}$$

The final output is computed as

$$o_i = \pi_{i,\text{cons}} \, o_i^{\text{cons}} + \pi_{i,\text{comp}} \, o_i^{\text{comp}} + \pi_{i,\text{conf}} \, o_i^{\text{conf}}. \tag{16}$$

## D.3 Aligned full-family comparison on the primary metric

Table 12 reports the aligned full-family comparison using the primary metric from the main text: binary F1 for CMU-MOSI and CMU-MOSEI, and Weighted F1 for MIntRec. The table includes EAMC, SEER-L0, SEER-L1, and the supplementary SEER-L2–L4 extensions under the same local evaluation pipeline. The

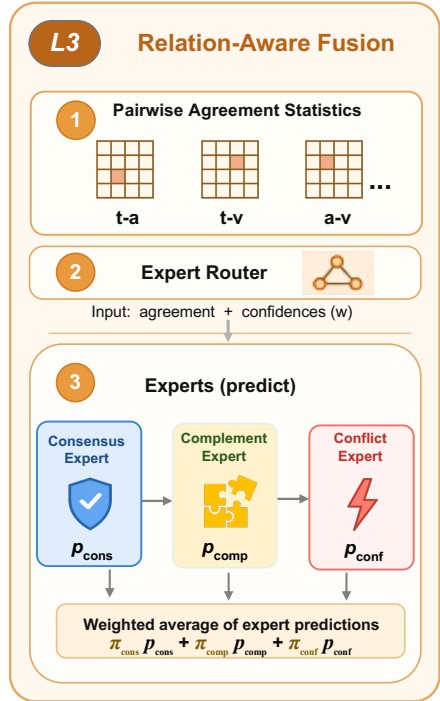

Figure 2: Supplementary architecture of SEER-L3. Pairwise agreement statistics between normalized modality representations and modality-confidence scores are used to predict mixture weights over consensus, complement, and conflict experts. SEER-L3 is reported as an appendix extension and is not the main routing variant used for the paper's primary claims.

Table 12: Aligned full-family comparison on the primary metric. CMU-MOSI and CMU-MOSEI report binary F1; MIntRec reports Weighted F1. Bold indicates the best local result within each dataset.

| Model | CMU-MOSI F1 ↑ | CMU-MOSEI F1 ↑ | MIntRec Weighted F1 ↑ |
|---|---|---|---|
| EAMC | $85.58 \pm 0.37$ | $85.33 \pm 0.22$ | $73.01 \pm 0.69$ |
| SEER-L0 | $85.49 \pm 0.41$ | $85.52 \pm 0.05$ | $72.51 \pm 0.85$ |
| SEER-L1 | $\mathbf{85.96 \pm 0.58}$ | $\mathbf{85.63 \pm 0.34}$ | $\mathbf{73.03 \pm 1.46}$ |
| SEER-L2 | $84.50 \pm 0.75$ | $85.43 \pm 0.20$ | $69.25 \pm 1.20$ |
| SEER-L3 | $85.62 \pm 0.02$ | $85.62 \pm 0.25$ | $72.24 \pm 0.73$ |
| SEER-L4 | $85.61 \pm 0.53$ | $85.54 \pm 0.10$ | $72.06 \pm 0.72$ |

purpose of this table is to document the broader method family, while the main paper focuses on EAMC, SEER-L0, and SEER-L1 as the controlled routing comparison.

Across the aligned benchmarks, SEER-L1 gives the highest mean primary metric among the local variants. SEER-L3 and SEER-L4 remain competitive on the sentiment datasets, but they do not consistently improve over SEER-L1, and SEER-L2 is weaker on MIntRec. These results support the main-text focus on SEER-L1 as the primary label-structured routing variant, while treating SEER-L2–L4 as supplementary extensions that broaden the design space.

### D.4 Supplementary results on unaligned CMU-MOSI

Table 13 reports supplementary results on unaligned CMU-MOSI. This setting is included to document performance beyond the aligned primary comparison, but it is not used as the main basis for the controlled

routing claims. The best local values are distributed across EAMC, SEER-L0, SEER-L1, and SEER-L4, indicating that the unaligned setting is more metric-dependent than the aligned comparison.

Table 13: Supplementary results on unaligned CMU-MOSI. Rows above the separator are published/reference results, while rows below are local results. Bold indicates the best local result.

| Method | Acc-7 ↑ | Acc-2 ↑ | F1 ↑ | MAE ↓ |
|---|---|---|---|---|
| EMOE (published) | 47.80 | 85.40 | 85.30 | 0.6970 |
| EAMC (ours) | 45.43 ± 1.09 | **86.18 ± 0.92** | 86.04 ± 0.93 | 0.7213 ± 0.0158 |
| SEER-L0 (ours) | 45.14 ± 0.42 | 86.13 ± 0.70 | **86.07 ± 0.71** | 0.7112 ± 0.0081 |
| SEER-L1 (ours) | **46.94 ± 1.54** | 85.57 ± 0.75 | 85.53 ± 0.73 | 0.7173 ± 0.0140 |
| SEER-L2 (ours) | 33.53 ± 15.69 | 85.06 ± 0.27 | 85.00 ± 0.33 | 0.9332 ± 0.2890 |
| SEER-L3 (ours) | 45.53 ± 0.37 | 85.57 ± 0.23 | 85.52 ± 0.19 | 0.7231 ± 0.0055 |
| SEER-L4 (ours) | 46.89 ± 1.11 | 85.93 ± 0.58 | 85.83 ± 0.58 | **0.6992 ± 0.0169** |

The unaligned CMU-MOSI results show that no single local variant dominates across all metrics. SEER-L1 gives the highest local Acc-7, SEER-L0 gives the highest local F1, EAMC gives the highest local Acc-2, and SEER-L4 gives the lowest MAE. This supports treating the unaligned setting as supplementary evidence rather than as the primary controlled comparison.

## D.5 Supplementary results on unaligned CMU-MOSEI

Table 14 reports supplementary results on unaligned CMU-MOSEI. This setting is included to document performance beyond the aligned primary comparison, but it is not used as the main basis for the controlled routing claims. Among the local variants, SEER-L3 gives the highest Acc-7, Acc-2, and F1, while SEER-L0 gives the lowest MAE. These results show that the supplementary extensions can remain competitive in the unaligned setting, while the main paper focuses on the aligned EAMC–SEER-L1 comparison.

Table 14: Supplementary results on unaligned CMU-MOSEI. Rows above the separator are published/reference results, while rows below are local results. Bold indicates the best local result.

| Method | Acc-7 ↑ | Acc-2 ↑ | F1 ↑ | MAE ↓ |
|---|---|---|---|---|
| EMOE (published) | 53.90 | 85.50 | 85.50 | 0.5300 |
| EAMC (ours) | 52.51 ± 0.64 | 85.56 ± 0.17 | 85.47 ± 0.18 | 0.5430 ± 0.0107 |
| SEER-L0 (ours) | 53.15 ± 0.43 | 85.69 ± 0.25 | 85.65 ± 0.24 | **0.5380 ± 0.0032** |
| SEER-L1 (ours) | 52.90 ± 0.36 | 85.73 ± 0.02 | 85.69 ± 0.02 | 0.5395 ± 0.0050 |
| SEER-L2 (ours) | 50.72 ± 1.72 | 85.06 ± 0.40 | 84.77 ± 0.69 | 0.5605 ± 0.0208 |
| SEER-L3 (ours) | **53.27 ± 0.32** | **85.99 ± 0.28** | **85.92 ± 0.28** | 0.5367 ± 0.0060 |
| SEER-L4 (ours) | 52.48 ± 1.23 | 85.64 ± 0.22 | 85.55 ± 0.16 | 0.5432 ± 0.0107 |

The unaligned CMU-MOSEI results show a different pattern from the aligned main comparison. SEER-L3 performs best among the local variants on Acc-7, Acc-2, and F1, suggesting that relation-aware expert fusion may be useful when modality alignment is less direct. However, because SEER-L3 changes the fusion stage in addition to the routing mechanism, these results are treated as supplementary rather than as evidence for the main label-structured routing claim.

## D.6 Supplementary full-family results on MIntRec

Table 15 reports the full local method-family comparison on MIntRec. Published baselines are included as reference points, while the local rows include reproduced EMOE, EAMC, SEER-L0, SEER-L1, and the supplementary SEER-L2–L4 extensions. Among the local variants, SEER-L1 gives the highest Weighted F1 and precision, while EAMC gives the highest accuracy and recall. The supplementary extensions remain competitive but do not improve the primary Weighted F1 metric over SEER-L1.

Table 15: Supplementary full-family results on MIntRec. Rows above the separator are published/reference results, while rows below are local results. Bold indicates the best local result.

| Method | Acc ↑ | Weighted F1 ↑ | Precision ↑ | Recall ↑ |
|---|---|---|---|---|
| MAG-BERT | 70.34 | 68.19 | 68.31 | 69.36 |
| MulT | 72.58 | 69.36 | 70.73 | 69.47 |
| MISA | 72.36 | 70.57 | 71.24 | 70.41 |
| EMOE (published) | 72.58 | 70.73 | 72.08 | 70.86 |
| EMOE (reproduced) | $72.36 \pm 1.36$ | $72.32 \pm 1.64$ | $73.10 \pm 1.87$ | $72.36 \pm 1.36$ |
| EAMC (ours) | $\mathbf{72.88 \pm 0.72}$ | $73.01 \pm 0.69$ | $74.11 \pm 0.93$ | $\mathbf{72.88 \pm 0.72}$ |
| SEER-L0 (ours) | $72.73 \pm 0.57$ | $72.51 \pm 0.85$ | $73.25 \pm 1.38$ | $72.73 \pm 0.57$ |
| SEER-L1 (ours) | $72.73 \pm 1.69$ | $\mathbf{73.03 \pm 1.46}$ | $\mathbf{74.23 \pm 1.03}$ | $72.73 \pm 1.69$ |
| SEER-L2 (ours) | $69.51 \pm 1.28$ | $69.25 \pm 1.20$ | $70.95 \pm 0.52$ | $69.51 \pm 1.28$ |
| SEER-L3 (ours) | $72.51 \pm 0.26$ | $72.24 \pm 0.73$ | $72.90 \pm 1.42$ | $72.51 \pm 0.26$ |
| SEER-L4 (ours) | $72.06 \pm 1.11$ | $72.06 \pm 0.72$ | $73.03 \pm 1.23$ | $72.06 \pm 1.11$ |

The MIntRec results support the main-text interpretation that SEER-L1 is the strongest local variant on the primary Weighted F1 metric, while EAMC remains highly competitive and gives the best local accuracy and recall. SEER-L2 is weaker on this dataset, suggesting that prototype-attentive temporal evidence extraction is not beneficial in the current MIntRec setting. SEER-L3 and SEER-L4 remain close to the reproduced EMOE baseline but do not improve over EAMC or SEER-L1. Therefore, the full-family MIntRec comparison supports keeping SEER-L1 as the primary label-structured routing variant and treating SEER-L2–L4 as supplementary extensions.

### D.7 Qualitative representation analysis

Figure 3 provides an illustrative visualization of fused representations on the aligned CMU-MOSI test split. We use a shared low-dimensional projection fitted on the same test examples across EAMC, SEER-L0, and SEER-L1 (van der Maaten & Hinton, 2008). Points denote individual utterances colored by sentiment value, and the overlaid centroid path summarizes coarse sentiment bins from negative to positive.

We treat this visualization as qualitative support only. It is intended to illustrate how the fused representation geometry changes across the controlled routing variants, not to serve as the main evidence for performance or routing reliability. The main evidence for the routing and prototype mechanisms comes from the quantitative ablations, statistical tests, missing-modality analysis, and routing diagnostics in Sections 4.3–4.5.

## E Supplementary Parameter-Efficiency Visualization

The main text reports exact trainable parameter counts in Table 16. Figure 4 provides a complementary visualization of the relationship between primary-metric performance and non-BERT trainable parameter count. The numerical table remains the primary reference for parameter counts.

As discussed in Section 4.6, all counts are generated using the same local counting rule. Total trainable parameters include the fine-tuned BERT text encoder, while non-BERT trainable parameters exclude only the BERT encoder. The MIntRec EMOE point should be interpreted as specific to the reproduced EMOE implementation and local feature configuration, where the raw-feature router scales substantially with input dimensionality.

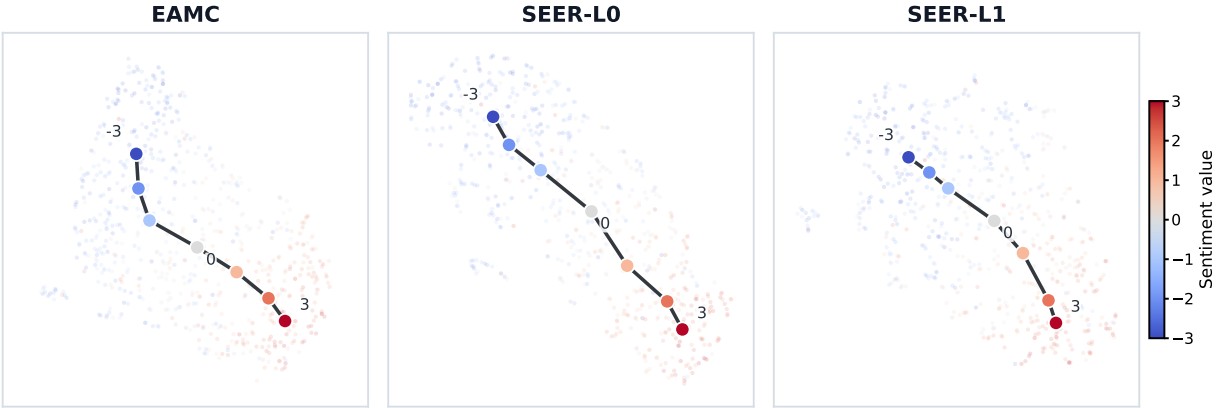

Figure 3: Qualitative comparison of fused representations on aligned CMU-MOSI. Points denote fused test representations for EAMC, SEER-L0, and SEER-L1 under a shared low-dimensional projection fitted on the same test examples. Colors indicate sentiment values, and the overlaid centroid path summarizes coarse sentiment bins from negative to positive. This visualization is illustrative; quantitative ablations and diagnostics provide the main evidence for the routing mechanism.

Table 16: Trainable parameter counts by dataset, reported in millions. Counts are based on the local implementations used in the experiments and are generated with the same counting rule. Total trainable parameters include the fine-tuned BERT text encoder; Non-BERT trainable parameters exclude only the BERT encoder.

| Dataset | EMOE | | EAMC | | SEER-L1 | |
|---|---|---|---|---|---|---|
| | Total | Non-BERT | Total | Non-BERT | Total | Non-BERT |
| MOSI | 317.3 | 207.8 | 120.9 | 11.4 | 120.7 | 11.2 |
| MOSEI | 361.1 | 251.7 | 120.9 | 11.4 | 120.7 | 11.2 |
| MIntRec | 1123.8 | 1014.3 | 120.4 | 10.9 | 120.2 | 10.7 |

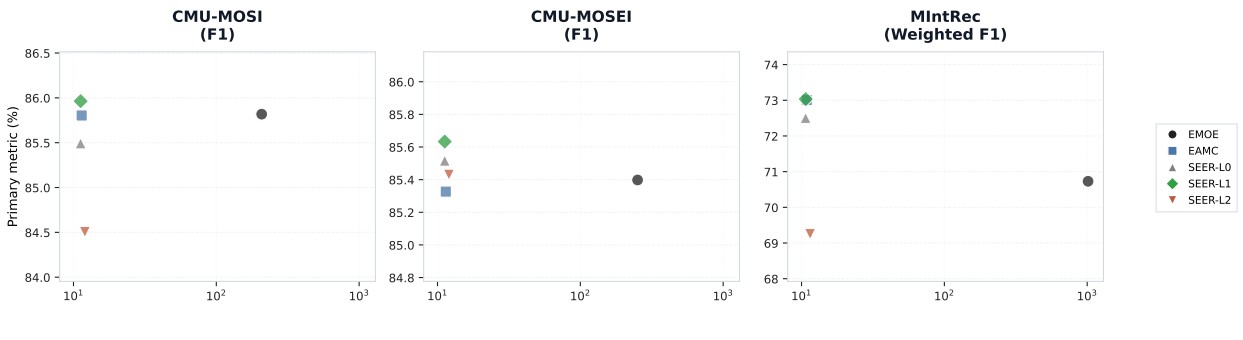

Figure 4: Supplementary performance–efficiency comparison. Each panel plots the primary F1-style metric against non-BERT trainable parameters on a log scale. The numerical parameter counts in Table 16 are the primary reference. The MIntRec EMOE point is configuration-specific, as described in Section 4.6.

