# OpenReview forum: "SEER: Label-Structured Modality Routing for Multimodal Sentiment Analysis and Intent Recognition"
_TMLR — Under review for TMLR_

### Review · Reviewer_7P3q · 2026-06-27

**Summary Of Contributions:**

Since many prior work in this field estimate modality reliability from raw input features, this paper first addresses this limitation through Emotion-Aware Modality Calibration (EAMC), which computes modality reliability using encoded modality representations, specifically the final hidden state of each modality. EAMC calculates a confidence score as the maximum soft assignment between each final hidden state and a learnable prototype bank, and then uses these confidence scores to construct the fused representation. However, this approach has two limitations. First, the encoded modality representations may not be sufficiently label-aware. Second, the prototype vectors may not reflect the downstream label structure because they are shared across modalities without explicit label alignment. To address the first issue, the authors introduce SEER-L0, which adds a label-aware contrastive loss. To address the second issue, they introduce SEER-L1, which uses both shared label-structured anchors and modality-specific private anchors. This allows the prototype structure to capture both label information and modality-specific style. The authors evaluate the proposed methods on CMU-MOSI, CMU-MOSEI, and MIntRec, and validate their effectiveness through empirical experiments. Overall, the paper claims that modality confidence should be estimated in a label-grounded semantic space.

## **Strengths**

- The paper tackles a meaningful problem: how modality weights should be estimated in adaptive multimodal fusion. Although label-related information is highly important for determining the reliability of each modality, prior work does not directly estimate which modality provides the most meaningful label-relevant evidence. This paper addresses this problem by introducing EAMC and the SEER variants.
- Furthermore, although the performance is only comparable to, and sometimes slightly lower than, the current SOTA baseline EMOE, the proposed methods still achieve reasonably strong performance while using substantially fewer non-BERT trainable parameters.

## **Weaknesses**

- The paper is somewhat difficult to follow because too many variants are introduced. It would improve readability if the authors presented one main method, perhaps SEER-L1, as the central contribution, and treated the remaining variants mainly as ablations. In the current version, EAMC, SEER-L0, SEER-L1, SEER-L2, SEER-L3, and SEER-L4 are all discussed, which makes the method section feel less focused. Even though SEER-L2, SEER-L3, and SEER-L4 are placed in the appendix, introducing this many variants feels excessive, especially because they do not consistently improve performance.
- Although the motivation for each component is somewhat understandable, the overall method appears overly complex. In particular, the training objective consists of four different terms, $L_{\text{task}} + \lambda_{\text{con}} L_{\text{con}} + \lambda_{\text{proto}} L_{\text{proto}} + L_{\text{aux}}$, which makes it unclear how much each component contributes to the final performance.
- It is unclear why the proposed methods are not directly compared against the existing baselines under the same experimental setting. In the main result tables, bolding is applied only among the authors’ own methods, while previously published baselines are not treated as direct competitors. Although the authors justify this by noting that the baselines use different reporting conditions, a more thorough direct comparison with these baselines would make the empirical claims much stronger.

**Audience:**

Yes

**Audience Explanation:**

The paper investigates an interesting question in multimodal learning: whether modality confidence should be estimated in a label-grounded semantic space rather than from raw features or generic prototype matching. This perspective is relevant to researchers working on multimodal fusion and adaptive routing, and the proposed framework provides a useful direction for incorporating label structure into modality confidence estimation. Although I believe the empirical evidence is not yet strong enough to fully support the proposed approach, the central research question and the proposed perspective are likely to be of interest to part of the TMLR community.

**Claims And Evidence:**

No

**Claims Explanation:**

- The main motivation, estimating modality scores using label-aware semantic information, is reasonable. However, the proposed method is not strongly supported by the empirical results. In particular, compared with EMOE, which appears to be the current SOTA baseline, the proposed methods are worse on several metrics.
- The qualitative analysis in Section 4.5 is also not fully convincing. The authors argue that SEER-L1 produces representations that are better separated along the sentiment axis, but EAMC and SEER-L0 also appear reasonably well separated in Figure 3. Moreover, the paper does not provide sufficient comparison with other prior routing methods that do not explicitly use label information, making it difficult to justify that label-aware routing is the key reason for the observed performance, even from a qualitative perspective.
- Given the complexity of the proposed method, a more detailed ablation study is necessary. The paper should clearly show why each component is needed and whether removing each component degrades performance. In particular, the training objective introduces additional terms such as the label-aware contrastive loss and the prototype-supervision loss, but there is no ablation that isolates the contribution of these losses. Without such analysis, it is difficult to determine whether the improvements come from the proposed label-structured routing mechanism or from other auxiliary training effects.

**Requested Changes:**

### **Critical**

- As mentioned in the summary, I believe the paper would benefit from focusing on a single main method that clearly represents the core contribution of the paper. The remaining variants could be presented primarily as ablations or extensions, which would significantly improve readability and help readers better understand the main message.
- The paper does not clearly explain how the prototype vectors are constructed and optimized. The authors should explicitly describe how the prototypes are initialized, updated during training, and supervised.
- More thorough ablation studies are necessary. In particular, the contribution of each loss term should be evaluated by removing them individually (e.g., w/o $L_{\text{con}}$, w/o $L_{\text{proto}}$). Furthermore, $L_{\text{aux}}$ is introduced in the objective but is never explicitly defined, making it difficult to understand the complete training procedure.
- The proposed method should be compared more directly with existing baselines under a fair experimental protocol. Although the authors explain that different reporting protocols prevent direct comparison, I encourage them to reproduce the strongest baselines under the same evaluation setting or evaluate their method under the baselines' protocol to provide a fair comparison.
- The paper would benefit from additional analysis explaining why SEER-L1 consistently improves the F1 metric. Simply reporting better performance is less informative without understanding which aspect of the proposed routing mechanism leads to the improvement.
- The qualitative representation analysis in Figure 3 is not sufficiently convincing. The differences between EAMC, SEER-L0, and SEER-L1 appear relatively subtle, making it difficult to support the claimed improvement in representation quality. Including prior routing baselines in the visualization would provide a more convincing justification for the proposed method.

### **Minor**

- It would be helpful to explicitly indicate learnable parameters throughout the paper for improved clarity. For example, the encoders could be written as $Enc_{\theta_m}$ (Page 4), the projection heads as $g_{\phi_m}$ (Page 6), and the prototype vectors could also be specified.
- On Page 4, the multimodal representation $u_i$ is referenced before it is formally defined.
- On Page 6, the role of $g_m$ is not clearly explained. Explicitly defining the module and its learnable parameters would improve readability.
- In Table 1, the best Acc-7 result is not boldfaced.
- Improving the readability of Figure 1 by simplifying the visual layout and enlarging key components would make the paper much easier to follow.

---

> ### Author Response · Authors · 2026-07-17
> **Response to Reviewer 7P3q**
>
> We thank the reviewer for the careful and constructive comments. The review helped us improve the paper’s focus, method clarity, and evidential support. In the revision, we refocused the main text around a controlled **EAMC → SEER-L0 → SEER-L1** progression, clarified the prototype-routing mechanism and training objective, added loss ablations, statistical tests, routing/prototype diagnostics, and missing-modality analysis, and moved supplementary variants and qualitative visualization to the appendix.
>
> **Concern 1: Too many variants make the method difficult to follow.**
>
> This concern led us to narrow the main manuscript. The main text now focuses on **EAMC**, **SEER-L0**, and **SEER-L1**: EAMC is the encoded-space routing foundation, SEER-L0 is a contrastive-only ablation, and SEER-L1 is the main label-structured routing refinement. This framing is introduced in the **Abstract** and **Introduction**, formalized in **Section 3**, and evaluated in **Section 4.3** and **Table 4**. SEER-L2, SEER-L3, and SEER-L4 are moved to **Appendix D** as supplementary extensions and are not used as primary evidence.
>
> **Concern 2: Prototype construction, optimization, supervision, and learnable parameters were unclear.**
>
> The revised **Method** section now makes these details explicit. **Section 3.2** explains the EAMC prototype bank initialization, normalization, and optimization. **Section 3.3** defines the SEER-L1 shared-private prototype structure, including modality-private banks, shared label anchors, classification/regression anchors, and end-to-end optimization. **Section 3.4** clarifies that encoders, prediction heads, projection heads, and prototype banks are learnable, while assignment vectors, confidence scores, and routing weights are sample-specific computed quantities.
>
> **Concern 3: The training objective and role of each loss term were not sufficiently clear.**
>
> The revised manuscript defines the fused task loss, unimodal auxiliary loss, label-aware contrastive loss, prototype-supervision loss, and shared regularization terms in **Section 3.4**, including loss coefficients for EAMC, SEER-L0, and SEER-L1. To evaluate the SEER-specific losses, **Table 6** reports a MIntRec ablation comparing full SEER-L1 with variants removing `$\\mathcal{L}_{\\mathrm{con}}$`, `$\\mathcal{L}_{\\mathrm{proto}}$`, or both. **Appendix B** adds a CMU-MOSEI ablation. These results show that the auxiliary losses act as dataset-dependent regularizers and structuring terms rather than uniformly dominant performance drivers.
>
> **Concern 4: The comparison with existing baselines, especially EMOE, should be clearer and fairer.**
>
> The revised protocol in **Section 4.1** separates published reference results from local protocol-equivalent comparisons. Published baselines are retained as external reference points because feature processing, seed selection, checkpoint selection, and metric implementations may differ. To strengthen the closest adaptive-routing comparison, we added a local reproduction of EMOE where available. **Tables 1–3** now include published baselines, reproduced EMOE, EAMC, and SEER-L1, and the captions clarify that boldface is applied only within local rows.
>
> **Concern 5: The empirical evidence should better explain SEER-L1 behavior, and the qualitative visualization was not sufficient.**
>
> The revised manuscript no longer relies on the qualitative visualization as central evidence; it is moved to **Appendix D.7** and described as illustrative only. The main text now provides quantitative mechanism-level analyses: **Table 5** reports paired bootstrap confidence intervals and permutation tests, **Table 6** reports loss ablations, **Table 7** reports routing/prototype diagnostics, and **Table 8** reports missing-modality analysis. These additions show that SEER-L1 confidence scores are empirically related to unimodal reliability and that prototype supervision improves shared-anchor alignment on ordinal sentiment data.
>
> **Concern 6: Some claims appeared stronger than the evidence supported.**
>
> We revised the claims throughout the **Abstract**, **Introduction**, **Section 4**, **Discussion and Limitations**, and **Conclusion**. The manuscript no longer describes SEER-L1 as producing uniform or statistically significant gains over EAMC. Instead, it frames EAMC as a strong encoded-space routing foundation and SEER-L1 as a label-structured refinement that preserves competitive primary-metric performance while making routing more structured and analyzable. This framing is supported by **Table 5**, where the confidence intervals overlap zero, and by the limitations discussion, which notes that the gains are modest and not statistically dominant.
>
> Overall, the revised manuscript is more focused, more transparent about the routing mechanism, and more cautious in its claims. We thank the reviewer again for the feedback, which directly shaped the revised version.

---

### Review · Reviewer_Byqu · 2026-07-13

**Summary Of Contributions:**

This paper studies adaptive modality routing for multimodal sentiment analysis and intent recognition, arguing that modality confidence should be estimated in a label-structured semantic space rather than from raw features or unconstrained prototypes. To study this, the paper introduces Emotion-Aware Modality Calibration (EAMC), a baseline that performs routing after semantic encoding. Building upon this, they propose Structured Evidence Estimation and Routing (SEER), where SEER-L0 adds a label-aware contrastive loss, SEER-L1 employs a shared-private prototype structure to match modality representations to label-anchored prototypes, and SEER-L2 adds a prototype-guided temporal evidence extraction. The experiments show that SEER-L1 outperforms EAMC baseline the most, demonstrating that estimating modality confidence with label-grounded semantic space helps adaptive multimodal routing.

**Audience:**

Yes

**Audience Explanation:**

See above.

**Claims And Evidence:**

No

**Claims Explanation:**

**Strengths**
1. The idea of shifting from raw-feature-level routing to an encoded label-structured space for modality routing is well-motivated and interesting.
2. Ablations on EAMC and different SEER variants demonstrate the contribution of different components.
3. Paper is well-written.

**Weaknesses**
1. The performance gains are marginal (Table 1-3) comparing to EAMC baseline, which cannot fully support the main claim that adaptive multimodal routing would be more effective when label-grounded semantic spaces are leveraged. It is also unclear whether
2. The model size (Table 5) analysis does not show strong positive results (i.e., SEER-L1 versus EAMC, where both approaches have almost the same number of parameters).
3. The evaluation results mainly focus on reporting downstream performance, though the main claim is that SEER estimates label-relevant modality confidence. There is little evidence that the learned weights actually correspond to reliable modality evidence. Some error analyses or experiments (e.g., missing or noisy modality) could be conducted.

**Requested Changes:**

See Weaknesses.

---

> ### Author Response · Authors · 2026-07-17
> **Response to Reviewer Byqu**
>
> We thank the reviewer for the constructive comments and for recognizing the motivation of moving modality routing from raw-feature space to an encoded, label-structured space. We revised the manuscript in response by calibrating the claims, adding statistical tests, adding routing/prototype diagnostics and missing-modality analysis, and clarifying the parameter-count interpretation.
>
> **Concern 1: The gains over EAMC are marginal and do not fully support a strong effectiveness claim.**
>
> We revised the claim framing throughout the **Abstract**, **Introduction**, **Sections 4.2–4.3**, **Discussion and Limitations**, and **Conclusion**. The paper no longer claims that SEER-L1 uniformly or significantly outperforms EAMC. Instead, EAMC is presented as a strong encoded-space routing foundation, while SEER-L1 is presented as a label-structured refinement that preserves competitive performance and makes routing more structured and analyzable.
>
> To support this calibrated interpretation, we added paired bootstrap confidence intervals and permutation tests in **Table 5**. These tests show dataset-dependent effects rather than significant dominance: the confidence intervals overlap zero, and **Section 4.3** now states that the observed margins should be interpreted cautiously. Thus, the revised claim is not that SEER-L1 is uniformly more effective than EAMC, but that it provides a more label-structured confidence-estimation mechanism while maintaining competitive task performance.
>
> **Concern 2: The parameter analysis does not show a strong advantage of SEER-L1 over EAMC.**
>
> We revised the parameter-efficiency discussion in **Section 4.6** and moved the detailed parameter table and visualization to **Appendix E**. The revised manuscript no longer presents SEER-L1 as having a meaningful parameter-count advantage over EAMC. Instead, it states that EAMC and SEER-L1 have similar parameter counts because they share the same backbone and differ mainly in the small prototype-routing structure.
>
> The parameter-count analysis is now interpreted more narrowly. In our reproduced EMOE implementation, the raw-feature router scales with flattened input dimensionality, whereas EAMC and SEER-L1 estimate confidence in encoded feature space. Therefore, the parameter result supports the compactness of encoded-space routing relative to this reproduced raw-feature expert-routing implementation, not a general advantage of SEER-L1 over EAMC.
>
> **Concern 3: Downstream performance alone does not show that SEER estimates reliable or label-relevant modality confidence.**
>
> This point motivated the main diagnostic additions in the revision. In **Section 4.5** and **Table 7**, we added routing-reliability diagnostics that compare the highest-weighted modality with the modality that has the lowest unimodal prediction error for each test sample. We also report Spearman correlation between routing weights and negative unimodal error. On aligned CMU-MOSEI, SEER-L1 selects the lowest-error modality in 53.82% of test samples, above the approximate 33.3% chance level, and shows positive rank correlation.
>
> We also added prototype-alignment diagnostics in **Table 7**. The full SEER-L1 model obtains 32.54% anchor accuracy and 1.069 anchor distance, while removing the prototype-supervision loss reduces anchor accuracy to 16.34% and increases anchor distance to 2.152. This supports the interpretation that prototype supervision improves shared-anchor alignment on ordinal sentiment data.
>
> Finally, we added missing-modality analysis in **Table 8** by removing text, audio, or video at test time. The results show that all variants remain strongly text-dependent, while removing audio or video has much smaller effects. We do not present this as evidence that SEER-L1 is uniformly more robust than EAMC; rather, it makes the modality dependence of the routing methods transparent.
>
> Overall, the revised manuscript adopts a more cautious position: SEER-L1 is not claimed to be uniformly superior to EAMC, but is presented as a label-structured routing refinement with competitive performance and stronger mechanism-level evidence for analyzable modality confidence.

---

### Review · Reviewer_BLLL · 2026-07-13

**Summary Of Contributions:**

The paper studies adaptive modality routing for multimodal sentiment analysis and intent recognition, focusing specifically on the representation space in which modality confidence is estimated before fusion. The authors introduce EAMC as a post-encoding routing baseline that moves reliability estimation from raw input features to encoded representations, then propose SEER, a staged extension that incorporates label structure into the confidence-estimation mechanism. SEER-L0 adds label-aware contrastive supervision with soft label-distance weighting for the regression setting, and SEER-L1 replaces unconstrained prototype matching with a shared-private anchor structure that grounds confidence scores in the downstream label space. SEER-L2, a prototype-attentive temporal evidence extension, is evaluated and reported as a negative result. Experiments on aligned CMU-MOSI, CMU-MOSEI, and MIntRec under a three-run multi-run protocol show modest improvements for SEER-L1 over EAMC on the primary F1-style metrics, alongside a substantial reduction in non-BERT trainable parameters relative to the reproduced adaptive-fusion baseline.

I found the paper quite well-written and accessible (as someone whose primary field is not sentiment analysis). The paper is also rather honest in its contributions, with SEER-L0's mixed effects and SEER-L2's failure documented clearly. The soft label-distance contrastive weighting [Eq. 3] is an interesting design choice for the ordinal sentiment setting that is underaddressed in the broader contrastive learning literature. The staged ablation design is methodologically clean and rigorous.

**Additional Comments:**

Again, this paper does not fall in my primary area,. I am reasonably confident in the statistical and experimental design concerns raised above, but I would weight a reviewer with closer familiarity with adaptive fusion architectures and multimodal affective computing more heavily. I may also values the cognitive-scientific motivations higher than what's typically expected for this topic.

**Audience:**

Yes

**Audience Explanation:**

Yes. Adaptive routing and confidence-weighted fusion for multimodal affective computing are active research directions, and the framing of modality confidence estimation as a representation-space design problem is a useful reorientation of how this problem is typically approached. The staged ablation is instructive even when the gains are small. The parameter efficiency result, though implementation-specific, is also worth knowing. The paper's contribution is incremental rather than transformative, but incremental and carefully executed work with honest negative results as such is contributive.

**Broader Impact Concerns:**

Yes, but perhaps none beyond what would be standard for a multimodal affective computing paper (the general risks of affective surveillance, etc).

**Claims And Evidence:**

No

**Claims Explanation:**

Partially, with critical caveats.The claims are accurately scoped and the experimental design is thoughtful, but the primary empirical claim, that SEER-L1 provides the most consistent improvement over EAMC on the primary F1-style metrics, is not adequately supported by the evidence as reported. The improvements on the two most interpretable comparisons fall well within one standard deviation of the baseline: SEER-L1 achieves 85.96 ± 0.58 vs. EAMC 85.80 ± 0.74 on MOSI, and 73.03 ± 1.46 vs. 73.01 ± 0.69 on MIntRec. No significance tests are reported anywhere in the paper, so it is not possible to determine whether any of the observed gains exceed what would be expected from sampling variance alone across three runs. Using "consistent improvement" to describe differences of this magnitude may be a bit overclaiming.

A secondary evidential gap concerns the theoretical narrative. SEER-L0 adds label-aware contrastive supervision directly to the routing representation space. Yet SEER-L0 does not reliably help and occasionally hurts. The paper notes this but does not investigate it, leaving open whether SEER-L1's gains derive from label-grounding specifically or from the shared-private architecture's inductive bias, which is a separate and potentially more mundane effect. An ablation separating these two components of SEER-L1 would address this. It would also help to bring in more substantial theoretical discussions particularly targeting affective computing and perhaps even alignments with human affective sciences. The current discussions focused primarily on the general multimodality literature, which I found to be underselling the design’s (cognitive-) scientific motivations in contrast to engineering choices, concerning its topics.

The evaluation protocol asymmetry also warrants more direct treatment. The local EAMC already exceeds the published EMOE on MIntRec under a more conservative multi-run protocol, which could reflect genuine improvement, a reproduced baseline that differs from the published one, or the single-run peak convention inflating the published number. The paper acknowledges this in passing but does not quantify it.

**Requested Changes:**

Critical:
1. Statistical significance testing is required before the claim of "consistent improvement" can be accepted. Bootstrap confidence intervals or permutation tests on the EAMC vs. SEER-L1 primary metric comparisons, across the three runs already completed, would be sufficient. If the tests do not support significance, the framing should be revised accordingly, e.g., "comparable performance with a more label-structured routing space”.

2. The design choice needs to be further explained in its principality within the affective science domains. The SEER-L0 failure also warrants further theoretical treatment, e.g. label-aware contrastive supervision does not improve routing despite directly targeting the identified limitation of EAMC, the paper should discuss why.

3. The evaluation protocol asymmetry between local multi-run aggregates and single-run published baselines should be addressed more directly. The paper should either reproduce all baselines under its own three-run protocol or explicitly quantify the expected direction and magnitude of the reporting gap, so that the headline comparisons with published numbers can be interpreted correctly.

Minor:

4. A more direct ablation of the shared-private structure versus the prototype supervision loss within SEER-L1 would clarify whether the gain comes from label-grounding or from the architectural factorization. This need not be a full new experiment.

5. Evaluation on at least one benchmark outside CMU-MOSI and CMU-MOSEI would strengthen generalizability claims. Both sentiment benchmarks are saturated to the point where within-family differences of under one F1 point are difficult to interpret reliably. MIntRec partially addresses this, but a broader evaluation would help.

---

> ### Author Response · Authors · 2026-07-17
> **Response to Reviewer BILL**
>
> We thank the reviewer for the careful and constructive assessment. We revised the manuscript in response by adding statistical tests, calibrating the claim framing, clarifying the role of SEER-L0, addressing protocol asymmetry, and adding diagnostic analyses.
>
> **Concern 1: The gains are marginal and should not be presented as consistent or statistically established improvements.**
>
> We revised the framing throughout the **Abstract**, **Introduction**, **Sections 4.2–4.3**, **Discussion and Limitations**, and **Conclusion**. SEER-L1 is no longer presented as uniformly or statistically dominant. Instead, EAMC is framed as a strong encoded-space routing foundation, while SEER-L1 is framed as a label-structured refinement that preserves competitive performance and makes routing more structured and analyzable.
>
> We also added paired bootstrap confidence intervals and permutation tests in **Table 5**. These tests show dataset-dependent effects rather than significant dominance: the confidence intervals overlap zero, and **Section 4.3** states that the margins should be interpreted cautiously.
>
> **Concern 2: SEER-L0 does not reliably help, so the role of label-aware contrastive supervision needs clearer interpretation.**
>
> The revised manuscript treats SEER-L0 as a diagnostic ablation rather than as a co-main method. In **Section 3.3**, SEER-L0 keeps the EAMC router unchanged while adding label-aware contrastive supervision. This tests whether label-aware representation structuring alone is sufficient when the confidence-estimation mechanism remains unchanged.
>
> The controlled comparison in **Section 4.3** and **Table 4** shows mixed changes for SEER-L0 relative to EAMC. We now interpret this as evidence that label-aware contrastive geometry alone does not guarantee stronger routing; the confidence-estimation mechanism itself also matters. This motivates SEER-L1’s shared-private label-structured routing. The loss ablations in **Table 6** and **Appendix B** further separate the effects of contrastive and prototype-supervision losses.
>
> **Concern 3: The comparison with published baselines may not be protocol-equivalent.**
>
> **Section 4.1** now explicitly separates published reference results from local protocol-equivalent comparisons. Published baselines are retained as external references because feature processing, seed selection, checkpoint selection, and metric implementation may differ. Direct comparisons focus on local rows evaluated under the same feature pipeline, checkpoint-selection rule, and metric implementation.
>
> We also added a local reproduction of EMOE as the closest adaptive-routing baseline where available. **Tables 1–3** include both published/reference rows and local rows, and the captions specify that boldface is applied only within local rows. **Appendix A** summarizes the local protocol and seeds.
>
> **Concern 4: The method’s added value should be supported beyond aggregate benchmark metrics.**
>
> We added mechanism-level analyses in **Section 4.5**. **Table 7** reports routing and prototype diagnostics, including top-route accuracy, Spearman correlation with negative unimodal error, anchor accuracy, and anchor distance. **Table 8** reports missing-modality analysis.
>
> These analyses show that routing weights are empirically related to unimodal reliability, prototype supervision improves shared-anchor alignment on ordinal sentiment data, and all variants remain strongly text-dependent. The manuscript also clarifies that routing weights are not causal explanations.
>
> **Concern 5: Affective-science motivation, limitations, and generalizability should be clearer.**
>
> The revised framing emphasizes modality reliability as label-relevant affective evidence rather than generic feature confidence. The **Introduction** clarifies that routing scores should reflect task-relevant evidence for the current sample. **Discussion and Limitations** further clarifies that routing weights are not causal or psychological explanations of emotion and that broader multilingual, asynchronous, noisy, and naturally corrupted settings remain future work. MIntRec is retained as a non-CMU benchmark, while broader evaluation is acknowledged as future work.
>
> Overall, SEER-L1 is now presented as a label-structured routing refinement with competitive performance and mechanism-level interpretability, while the statistical tests and limitations make clear that the current evidence does not establish significant dominance over EAMC.

---

### Author Response · Authors · 2026-07-17
**Author Revision Summary and Point-by-Point Responses**

We thank the reviewers and the Action Editor for the careful and constructive feedback. We have uploaded a revised manuscript that substantially refines the paper’s scope, method presentation, experimental evidence, and claim framing.

The main changes are summarized below.

1. We refocused the manuscript around a controlled EAMC → SEER-L0 → SEER-L1 progression. EAMC is now presented as the encoded-space routing foundation, SEER-L0 as a contrastive-only ablation, and SEER-L1 as the main label-structured routing refinement. Additional variants are moved to the appendix as supplementary extensions.

2. We clarified the method description, including prototype initialization, normalization, optimization, learnable parameters, routing weights, projection heads, shared/private prototype banks, and the complete training objective.

3. We added controlled ablations and statistical tests. In particular, we report paired bootstrap confidence intervals and permutation tests for SEER-L1 versus EAMC on the primary metric. The revised manuscript now explicitly frames the results as dataset-dependent effects rather than statistically significant dominance over EAMC.

4. We added routing and prototype diagnostics to examine whether learned routing weights correspond to modality reliability and whether prototype supervision improves shared-anchor alignment. We also added missing-modality analysis to make the modality dependence of the routing methods transparent.

5. We added a local reproduction of EMOE as the closest adaptive-routing baseline where available. Published baselines are retained as external reference points, and the manuscript now clearly distinguishes local protocol-equivalent comparisons from published single-model results.

6. We revised the discussion, limitations, and conclusion to avoid overclaiming. The revised manuscript now presents SEER-L1 as a structured and analyzable refinement of encoded-space routing, rather than as a uniformly dominant performance model.

We provide point-by-point responses to each reviewer below.